# USP8 Down-Regulation Promotes Parkin-Independent Mitophagy in the *Drosophila* Brain and in Human Neurons

**DOI:** 10.3390/cells12081143

**Published:** 2023-04-13

**Authors:** Sofia Mauri, Greta Bernardo, Aitor Martinez, Mariavittoria Favaro, Marta Trevisan, Gael Cobraiville, Marianne Fillet, Federico Caicci, Alexander J. Whitworth, Elena Ziviani

**Affiliations:** 1Department of Biology, University of Padova, 35121 Padova, Italy; 2MRC Mitochondrial Biology Unit, Cambridge Biomedical Campus, University of Cambridge, Cambridge CB2 0XY, UK; 3Department of Molecular Medicine (DMM), University of Padova, 35121 Padova, Italy; 4Laboratory for the Analysis of Medicines, Center for Interdisciplinary Research on Medicines (CIRM), Quartier Hopital, University of Liege, Avenue Hippocrate 15, 4000 Liege, Belgium

**Keywords:** autophagy, mitophagy, Parkin, DUBs, USP8

## Abstract

Stress-induced mitophagy, a tightly regulated process that targets dysfunctional mitochondria for autophagy-dependent degradation, mainly relies on two proteins, PINK1 and Parkin, which genes are mutated in some forms of familiar Parkinson’s Disease (PD). Upon mitochondrial damage, the protein kinase PINK1 accumulates on the organelle surface where it controls the recruitment of the E3-ubiquitin ligase Parkin. On mitochondria, Parkin ubiquitinates a subset of mitochondrial-resident proteins located on the outer mitochondrial membrane, leading to the recruitment of downstream cytosolic autophagic adaptors and subsequent autophagosome formation. Importantly, PINK1/Parkin-independent mitophagy pathways also exist that can be counteracted by specific deubiquitinating enzymes (DUBs). Down-regulation of these specific DUBs can presumably enhance basal mitophagy and be beneficial in models in which the accumulation of defective mitochondria is implicated. Among these DUBs, USP8 is an interesting target because of its role in the endosomal pathway and autophagy and its beneficial effects, when inhibited, in models of neurodegeneration. Based on this, we evaluated autophagy and mitophagy levels when USP8 activity is altered. We used genetic approaches in *D. melanogaster* to measure autophagy and mitophagy in vivo and complementary in vitro approaches to investigate the molecular pathway that regulates mitophagy via USP8. We found an inverse correlation between basal mitophagy and USP8 levels, in that down-regulation of USP8 correlates with increased Parkin-independent mitophagy. These results suggest the existence of a yet uncharacterized mitophagic pathway that is inhibited by USP8.

## 1. Introduction

Loss of protein and organelle homeostasis is well documented during aging. However, whereas physiological decline in proteostasis is expected in older adults, this seems to be more severe and pathologically relevant in age-related neurodegenerative disorders, such as Parkinson’s disease (PD), Alzheimer’s disease (AD), Huntington’s disease (HD) and amyothrophic lateral sclerosis (ALS) [1].

When it comes to familiar PD in particular, the molecular link between deficient mechanisms of proteostasis and disease onset is more evident, in that two genes mutated in familiar forms of PD, Ser/Thr kinase PINK1 and E3-ubiquitin ligase Parkin, operate as key regulators of mitochondrial degradation. Under normal conditions, PINK1 levels are maintained low: the protein is imported into mitochondria through its mitochondrial targeting sequence, and it is processed by the matrix processing peptidase (MPP) and the presenilins-associated rhomboid-like (PARL) protease. Cleaved PINK1 is retrotranslocated to the cytosol and rapidly degraded by the proteasome. When mitochondria depolarize following mitochondrial damage, PINK1 fails to be imported into mitochondria and its cleavage is reduced. The protein accumulates on the outer mitochondrial membrane (OMM), and its stabilization leads to an increase in its kinase activity and autophosphorylation. PINK1 recruits cytosolic Parkin to the mitochondria by phosphorylating ubiquitin on serine 65 (Ser65) and Parkin. Parkin phosphorylation, which also occurs at Ser65, leads to the release of Parkin auto-inhibited conformation and promotes its interaction with phospho-ubiquitin. Activated Parkin polyubiquitinates itself and multiple substrates on the OMM, including VDAC, TOM20, FIS1, Miro and mitochondrial pro-fusion proteins Mfn1, Mfn2 and Marf (fly homologue of Mfn1/2). The ubiquitin chains formed on the proteins of the OMM serve as substrates for the kinase activity of PINK1, which in turn recruits more Parkin, leading to a feed-forward loop that culminates with the recruitment of autophagy receptors p62, Optineurin (OPTN) and nuclear dot protein 52 kDa (NDP52). These receptors interact with mitochondria via their ubiquitin-binding domain and with the autophagosome via their LC3-interacting region (LIR) motif, ensuring the targeting of mitochondria to the forming phagophore [2,3,4]. Importantly, in addition to Parkin, other E3-ubiquitin ligases, such as MUL1, SMURF1 and Gp78, have been proposed to ubiquitinate mitochondrial proteins and promote mitophagy in a Parkin-independent fashion. These alternative E3 ubiquitin ligases often operate on the same mitochondrial targets that are ubiquitinated by Parkin, as is the case for E3 ubiquitin ligase MUL1, which shares the same substrate (Mfn) with Parkin [5]. This evidence suggests that Parkin is not indispensable for mitophagy but rather acts to amplify PINK1 signal.

Besides ubiquitin-mediated mitophagy, mitochondrial removal can also be promoted by autophagy receptors, such as NIX, BNIP3, FUNDC1, Cardiolipin and Prohibitin 2 (PHB2). Mitophagy receptors can recruit autophagosomes to mitochondria in a PINK1/Parkin independent fashion [6], indicating that PINK1/Parkin-independent mitophagy pathways also exist, which can presumably be enhanced to ameliorate mitochondrial quality control in their absence.

In the quest of potential enhancers of Parkin-independent mitophagy pathways, deubiquitinating enzymes (DUBs) are interesting candidates for their activity on the ubiquitination status of proteins. In this respect, ubiquitin-specific protease USP8 is a compelling target for its reported role in the modulation of autophagy and mitophagy and its physiologically relevant catalytic and non-catalytic (scaffolding) activities. In particular, USP8 represents a typical multidomain DUB that exerts several functions: it deubiquitinates the epidermal growth factor receptor (EGFR) on the plasma membrane and prevents its degradation by the endosome–lysosome pathway, a process known as receptor down-regulation. As a result, USP8 activity enhances the stability of EGFR (an essential regulator of proliferation and differentiation), whereas USP8 inhibition promotes EGFR down-regulation. This is consistent with the antitumorigenic effects of USP8 inhibition that have been reported in several cancer models [7,8]. Another direct target of USP8 deubiquitinating activity is EPG5, an autophagy regulator that mediates autophagosome/lysosome fusion. EPG5 maintains a high autophagic flux to support ESCs stemness. USP8 deubiquitinates EPG5, an event that is required for EPG5-LC3 interaction, and plays an essential role in EPG5-dependent autophagy in the maintenance of ESCs stemness [9]. USP8 also plays a critical role in the development and homeostasis of T cells, in that specific inactivation of USP8 in T cells affects thymocyte maturation via specific activation of genes controlled by the transcription factor Foxo. As a consequence, specific ablation of USP8 in T cells profoundly affects the homeostasis and development of the immune system [10].

Besides the ubiquitin-specific protease activity, USP8 plays an essential role as a scaffolding protein that is connected to the endosomal trafficking and transport [11]. In particular, USP8 harbors an N-terminal microtubule interacting and transport (MIT) domain and two atypical central SH3-binding motifs (SH3BMs) that flank a 14-3-3 protein-binding motif (14-3-3BM). The MIT domain interacts with charged multivesicular body proteins (CHMP), components of the endosomal sorting complexes required for transport III (ESCRT-III), whereas the SH3BM interacts with the signal transducing adaptor molecule (STAM), which is a part of the ESCRT-0 complex [12].

As it can be inferred from these essential catalytic and non-catalytic functions of USP8 the expression of this DUB is an absolute requirement for proper tissue development and differentiation and for endosome sorting and trafficking. Not surprisingly, USP8 KO is embryonically lethal in mammals and flies, whereas mutations that enhance USP8 catalytic activity causes Cushing disease in humans by sustaining EGFR signaling.

Of particular relevance for this work, USP8 activity has been recently linked to autophagy and mitophagy, although with contrasting results. In particular, USP8 loss of function in *D. melanogaster* leads to the accumulation of autophagosomes due to a blockade of the autophagic flux [13,14], whereas in HeLa cells [13] and HEK293T cells [15], USP8 knockdown enhances the autophagic flux. More recently, it was reported that USP8 negatively regulates autophagy by deubiquitinating the autophagy factors TRAF6, BECN1 and p62 [16], supporting the hypothesis that USP8 reduction can be used to promote autophagy. USP8 is also directly connected to Parkin-mediated mitophagy by controlling the removal of K6-linked ubiquitin from Parkin. Stabilization of ubiquitin moieties on the Parkin molecule by USP8 knockdown does not seem to correlate to an increase in Parkin degradation. On the contrary, Parkin levels increase when USP8 is downregulated, whereas CCCP-induced Parkin recruitment is delayed, as well as mitophagy [17].

The observations that USP8 down-regulation might inhibit the autophagic flux and mitophagy points to a potential aggravating effect for USP8 reduction in models in which accumulation of misfolded proteins and aberrant mitochondria is implicated. Nevertheless, many publications indicate that USP8 down-regulation is protective in models that can benefit from enhanced proteostasis. In particular, USP8 knockdown decreases β-secretase levels and Aβ production in an in vitro model of AD [18]. USP8 knockdown also leads to increased lysosomal degradation of α-synuclein, and it protects from α-synuclein-induced toxicity and cell loss in an α-synuclein fly model of PD [19]. We also previously demonstrated that USP8 down-regulation or its pharmacological inhibition ameliorates the phenotype of PINK1 and Parkin KO flies by preventing neurodegeneration and rescues mitochondrial defects, lifespan and locomotor dysfunction in these flies [20]. In these models of neurodegeneration, in vitro and in vivo downregulation of USP8 correlated with decreased levels of mitochondrial fusion protein Marf/MFN and normalized Marf/MFN levels of PINK1 KO flies that are pathologically elevated. 

In this work, we wanted to extend the work that has been performed and explore the effect of USP8 down-regulation in neurons in the context of autophagy and mitophagy, using *D. melanogaster* as a model organism. We subsequently investigated the effect of USP8 inhibition using the potent inhibitor DUBs-IN-2 [21] in mammalian cells, particularly in iNeurons generated from human embryonic stem cells (hESCs) [22]. We found that USP8 down-regulation enhances autophagy and mitophagy in flies and in neurons of human origin, providing a mechanistic explanation for the protective effect of USP8 reduction observed in several models of neurodegeneration.

## 2. Materials and Methods

### 2.1. Fly Strains and Husbandry

Drosophila were raised under standard conditions at 25 °C with a 12 h:12 h light:dark cycle, on agar, cornmeal, yeast food. w^1118^ (BDSC_5905), UAS-GFP-mCherry-Atg8a (BDSC_37749) and nSybGAL4 (BDSC_51635) fly lines were obtained from the Bloomington Drosophila Stock Center. The UAS-USP8^GD1285^ RNAi and UAS Marf RNAi (VDRC 105261) lines were obtained from the VDRC Stock Center. The USP8^−/+^ line was kindly provided by Satoshi Goto [23]. The lines park^25^, UAS-mito-QC and Act5cGAL4 were generated previously [24,25]. For larval experiments, L3 wandering larvae were selected based on their phenotypes. 

### 2.2. Larvae Dissection and Fixation

Larval brain dissections were performed in PBS and fixed in 4% formaldehyde, pH 7.0 for 20 min. Subsequently, brains were washed in PBS and mounted in Mowiol^®^ 4–88 (Sigma-Aldrich, 81381, St. Louis, MO, USA). Samples were dissected in the morning or in the afternoon and imaged in the afternoon of the same day or the following morning, respectively.

### 2.3. Microscopy and Image Analysis

Fluorescence microscopy imaging was performed using a Zeiss LSM 900 confocal microscope equipped with 100× Plan Apochromat (oil immersion, NA 1.4) objective lenses at 2× digital zoom. Z-stacks were acquired at 0.5 µm steps. For each larval brain, two images of different areas were taken. In the graphs, each data point represents one brain. For both autophagy and mitophagy analyses, samples were imaged via sequential excitations (488 nm, green; 561 nm, red). Laser power and gain settings were adjusted depending on the fluorophore but were maintained across samples. For quantification, maximum intensity projections were created and analyzed using Fiji (ImageJ 2.9.0) software. For autolysosomes quantification, the number of mCherry-only puncta was quantified using the mQC-counter plugin [26], maintaining the same parameters across samples (radius for smoothing images = 1; ratio threshold = 0.8; red channel thresh: stdDev above mean = 1). To quantify autophagosomes (yellow dots), the green and red channels were threshold (mean intensity + 3 × StdDev and mean intensity + 2 × StdDev, respectively). Objects present in both the mCherry and GFP masks were counted. For mitolysosome quantification, the number of mCherry-only puncta was quantified using the mQC-counter plugin [26], maintaining the same parameters across samples (radius for smoothing images = 1; ratio threshold = 1; red channel thresh: stdDev above mean = 1).

### 2.4. Electron Microscopy

Thoraces were prepared from 3-day-old adult flies and fixed O.N. in 2% paraformaldehyde and 2.5% glutaraldehyde. After rinsing in 0.1 M cacodylate buffer with 1% tannic acid, samples were postfixed in 1:1 2% OsO4 and 0.2 M cacodylate buffer for 1 h. Samples were rinsed, dehydrated in an ethanol series and embedded by using Epon. Ultrathin sections were examined using a transmission electron microscope.

### 2.5. S2R+ Cell Culture

*D. melanogaster* S2R+ cells were cultured in Schneider’s Drosophila medium (Biowest, Nuaillé, France) supplemented with 10% heat-inactivated fetal calf serum. Cells were maintained at 25 °C and passaged routinely.

### 2.6. Gene Silencing 

Drosophila dsRNA probes were prepared using MEGA script kit (Ambion, Waltham, MA, USA) following the manufacturer’s instructions. The CG5798/USP8 dsRNA probe was acquired from the Sheffield RNAi Screening Facility. 

### 2.7. S2R+ Transfection, Cell Imaging Acquisition and Processing

A total of 2 × 10^5^ S2R+ cells were plated in a 24-well plate and transfected with 1 μg DNA, 1.5 μL Effectene (QIAGEN), 1.5 μL enhancer and 20 μL EC buffer 1 day after plating following the manufacturer’s instructions. When required, copper sulfate solution was added to the cells to induce plasmid expression. The following plasmids were used: UAS-mt-Keima, actin-GAL4, mito-dsRed and Parkin-GFP. For USP8 down-regulation, cells were treated with control or USP8 dsRNA probes (ctrl RNAi, Usp8 RNAi). For the Parkin recruitment experiment, after plating, S2R+ cells were treated with either 10 μM CCCP (Sigma-Aldrich) (treated cells) or equal amount of DMSO (control cells) for the indicated amount of time. Cells were collected for the experiments 72 h after transfection.

For imaging acquisition and processing, S2R+ cells were plated on 24 mm round glass coverslips and co-transfected with the indicated plasmids (Parkin-GFP, mito-DsRed, LC3-RFP) and/or dsRNA probes (ctrl RNAi, Usp8 RNAi) for 48–72 h before imaging. Images were acquired using a UPlanSApo 60×/1.35 NA objective (iMIC Adromeda, TILL Photonics, Pleasanton, CA, USA) upon excitation with 561 and 488 nm lasers. Parkin translocation was evaluated by counting the number of cells with Parkin puncta on mitochondria. Autophagosome–mitochondria interaction was evaluated by measuring the percentage of LC3 that co-localized with mitochondria by Mander’s coefficient of co-localization using ImageJ JACoP (“Just Another Colocalization Plug in”), following 3D volume rendered reconstruction of 40 *z*-axis images, separated by 0.2 μm (software: ImageJ, plug in: volume and JACoP).

### 2.8. Flow Cytometry

A total of 72 h after transfection, S2R+ cells were gently washed with PBS and collected in 300 μL HBSS + HEPES for flow cytometry. mt-Keima expressing cells were analysed by flow cytometry (BD FACSAria sorter, San Jose, CA, USA) to measure mitophagy levels in control cells (ctrl RNAi) or cells with altered USP8 expression (Usp8 RNAi), following an established protocol [12]. Briefly, cells were analysed with a flow cytometer (BD FACSAria™) equipped with 405 nm and 561 nm lasers. Cells were simultaneously excited with a violet laser (405 nm), with emission detected at 610 ± 10 nm with a BV605 detector, and with a yellow-green laser (561 nm), with emission detected at 610 ± 10 nm by a PE-CF594 detector. mt-Keima-positive cells were gated based on their ratio of emission at PE-CF594/BV605 in a “high” or “low” gate. The proportion of mitophagic cells was represented by the percentage of cells in the “high” gate among the mt-Keima-positive population.

### 2.9. Thermal Stability Assay

A total of 1 × 10^6^ cells were plated onto 10 cm Petri dishes and treated after 24 h with dsRNA probes (ctrl RNAi or Usp8 RNAi). Next, cells were resuspended in PBS, snap-frozen in liquid nitrogen and thawed 4 times. The solution was aliquoted into a PCR strip and incubated at the indicated temperature for 3 min. The lysates were centrifuged at 16,000× *g* for 30 min at 4 °C. The soluble fraction was loaded onto SDS-PAGE gel.

### 2.10. Protein Extraction

Cells were collected in lysis buffer (150 mM NaCl, 50 mM Tris-HCl, 1% NP-40, 0.25% sodium deoxycholate and 1 mM EDTA in distilled water, adjusted pH to 7.4) with freshly added protease inhibitor cocktail (PIC) and incubated on ice for 30 min before being centrifuged at maximum speed at 4 °C for 15 min. The protein concentrations of the samples was determined using the Pierce™ BCA Protein Assay Kit (ThermoFisher Scientific, Waltham, MA, USA). 2-Mercaptoethanol (Sigma-Aldrich) was mixed into the samples and the proteins were then denatured at 95 °C for 5 min.

### 2.11. Western Blot

Western blots were performed using ExpressPlus PAGE Gel 4–12% or 4–20% (GenScript, Piscataway, NJ, USA). Proteins were transferred to PVDF membranes (MERCK-Millipore) using the Trans-Blot Turbo Transfer System (Bio-Rad, Hercules, CA, USA) following manufacturer’s instructions. Membranes were incubated with indicated antibodies and imaged with ImageQuant LAS4000. Band densiometry quantification was performed using ImageJ software. The following antibodies were used: anti-Actin (1:1000; Chemicon MAB1501, Tokyo, Japan), α-ATP5A (1:4000, Abcam ab14748, Cambridge, MA, USA), α-Cyclophilin D (1:500, Abcam ab110324), α-TOM20 (1:1000, Santa Cruz sc-11415, Dallas, TX, USA) and α-VDAC (1:1000, Abcam ab15895). Canonical secondary antibodies used were sheep anti-mouse or donkey anti-rabbit HRP (GE Healthcare, Chicago, IL, USA). Immunoreactivity was visualized with Immobilon Forte Western HRP substrate (Millipore, Burlington, MA, USA).

### 2.12. Isolation and Identification of Ubiquitin Modifications by Mass Spectrometry

To identify the full repertoire of USP8 targets, protein lysates extracted from 200 CTR (Act5cGAL4/+) or USP8 KD (Act5cGAL4/+; UAS USP8 RNAi/+) flies were subjected to immunoaffinity isolation and mass spectrometry analysis to enrich and identify K-GG peptides from digested protein lysates as previously described [13]. Fly lysates were prepared in lysis buffer (9 M urea, 20 mM HEPES pH 8.0, 1 mM sodium orthovanadate, 2.5 mM sodium pyrophosphate and 1 mM β-glycerophosphate) by brief sonication on ice. Protein samples (20 mg) were reduced at 55 °C for 30 min in 4.1 mM DTT, cooled 10 min on ice and alkylated with 9.1 mM iodoacetamide for 15 min at room temperature in the dark. Samples were diluted 3-fold with 20 mM HEPES pH 8.0 and digested in 10 μg mL^−1^ trypsin-TPCK (Promega, Madison, WI, USA) overnight at room temperature. Following digestion, trifluoroacetic acid (TFA) was added to a final concentration of 1% to acidify the peptides before desalting on a Sep-PakC18 cartridge (Waters). Peptides were eluted from the cartridge in 40% acetonitrile and 0.1% TFA, flash frozen and lyophilized for 48 h. Dry peptides were gently resuspended in 1.4 mL 1× immunoaffinity purification (IAP) buffer (Cell Signaling Technology, Danvers, MA, USA) and cleared by centrifugation for 5 min at 10,000× *g* rcf at 4 °C. Precoupled anti-KGG beads (Cell Signaling Technology) were washed in 1× IAP buffer before contacting the digested peptides. Immunoaffinity enrichment was performed for 2 h at 4 °C. Beads were washed 2× with IAP buffer and 4× with PBS before 2× elution of peptides in 0.15% TFA for 10 min each at room temperature.

### 2.13. LC-Chip-MS/MS Analysis

Chromatographic separation was performed on a chip including a 160 nL trapping column and a 150 mm length and 75 μm internal diameter analytical column, both packed with a Zorbax 300SB 5 μm C18 phase (Agilent Technologies, Waldbronn, Germany). The mobile phase was composed of H_2_O/FA (100:0.1, *v*/*v*) (A) ACN/H_2_O/FA (90:10:0.1, *v*/*v*/*v*) and (B) degassed by ultrasonication for 15 min before use. Analytical process was performed in two steps: First, the sample was loaded on the trapping column during an isocratic enrichment phase using the capillary pump delivering a mobile phase in isocratic mode composed of H_2_O/ACN/FA (97:3:0.1, *v*/*v*/*v*) at a flow rate of 4 µL/min. A flush volume of 6 µL was used to remove unretained components. Then, after valve switching, a gradient elution phase in backflush mode was performed through the enrichment and analytical columns using the nanopump. The analysis was performed using a gradient starting at 3% B that linearly ramped up to 45% B in 30 min at a flow rate of 300 nL/min and then up to 95% B in 5 min. The column was then rinsed with 95% B for 5 min before returning to 3% B. Ten column volumes were used for re-equilibration prior to the next injection. The total analysis time was 43 min for each run. All the experiments were carried out with an 8 µL sample injection volume. During the analysis, the injection needle was thoroughly rinsed three times from the inside and the outside with a mix of ACN/H_2_O/TFA (60:40:0.1, *v*/*v*/*v*) commanded by an injection program set in the injector parameters. The identifications were performed using an electrospray MS-MS using a 6340 series ion trap mass spectrometer (Agilent Technologies). The collision energy was set automatically depending on the mass of the precursor ion. Each MS full scan was followed by MS/MS scans of the six most intense precursor ions detected in the MS scan (exclusion time: 1 min). The results were subsequently introduced into the database for protein identification searches using Spectrum Mill (Agilent Technologies). All searches were carried out with “Drosophila melanogaster” as taxonomy in the NCBInr database and 0.5 Da of tolerance on MS/MS fragments. The search parameters allowed fixed modifications for cysteine (carboxyamidomethylation) and variable modifications for methionine (oxidation) and for lysine (ubiquitination). Two missed cleavages were allowed. VML score displays the VML (variable modification localization) score of the modification selected, which is the difference in score between equivalent identified sequences with different variable modification localizations. A VML score of >1.1 indicates confident localization. A score of 1 implies that there is a distinguishing ion of b or y ion type and 0.1 means that when unassigned, the peak is 10% of the intensity of the base peak.

### 2.14. Generation of Stable Mitophagic Flux Reporters hESC Lines and Differentiation

H9 hESCs (WiCell Institute) were cultured in TeSR™-E8™ medium (StemCell Technologies, Vancouver, BC, Canada) on Matrigel-coated tissue culture plates with daily medium change. Cells were passaged every 4–5 days with 0.5 mM EDTA in DMEM/F12 (Sigma). For introduction of TRE3G-NGN2 into the AAVS1 site, a donor plasmid pAAVS1-TRE3G-NGN2 was generated by replacing the EGFP sequence with an N-terminal flag-tagged human NGN2 cDNA sequence in plasmid pAAVS1-TRE3G-EGFP (Addgene plasmid # 52343, Watertown, MA, USA). A total of 5 μg of pAAVS1-TRE3G-NGN2, 2.5 mg hCas9 (Addgene plasmid # 41815) and 2.5 mg gRNA_AAVS1-T2 (Addgene plasmid # 41818) were electroporated into 1 × 10^6^ H9 cells. The cells were treated with 0.25 mg/mL puromycin for 7 days and surviving colonies were expanded and subjected to genotyping. H9 hESC harboring the mitochondrial matrix mCherry-GFP flux reporter were generated by transfection of 1 × 10^5^ cells with 1 μg pAC150-PiggyBac-matrix-mCherry-eGFPXL [24] and 1 μg pCMV-HypBAC-PiggyBac-Helper (Sanger Institute) in conjunction with the transfection reagent FuGENE HD (Promega, Madison, WI, USA). The cells were selected and maintained in TeSR™-E8™ medium supplemented with 200 mg/mL hygromycin; hygromycin was kept in the medium during differentiation to iNeurons. For H9 hESC conversion into iNeurons, cells were treated with Accutase (Thermo Fisher Scientific, Waltham, MA, USA) and plated on Matrigel-coated tissue plates in DMEM/F12 supplemented with 1× N2, 1× NEAA (Thermo Fisher Scientific), human brain-derived neurotrophic factor (BDNF, 10 ng/mL, PeproTech, Waltham, MA, USA), human neurotrophin-3 (NT-3, 10 ng/L, PeproTech), human recombinant laminin (0.2 mg/mL, Life Technologies, Waltham, MA, USA), Y-27632 (10 mM, PeproTech) and doxycycline (2 mg/mL, Sigma-Aldrich) on Day 0. On Day 1, Y-27632 was withdrawn. On Day 2, the medium was replaced with neurobasal medium supplemented with 1× B27 and 1× Glutamax (Thermo Fisher Scientific) containing BDNF, NT-3 and 2 mg/mL doxycycline. Starting on Day 4, half of the medium was replaced every other day thereafter. On Day 7, the cells were treated with Accutase (Thermo Fisher Scientific) and plated on Matrigel-coated tissue plates. Doxycycline was withdrawn on Day 10. Treatments and experiments were performed between days 11 and 13.

### 2.15. Statistical Analysis

Statistical analyses were performed using GraphPad Prism 8 software. Data are represented as box plots (min to max, all data points showed) or as mean ± SEM. Statistical significance was measured by an unpaired t-test, one-way or two-way ANOVA or Kruskal–Wallis nonparametric test followed by ad hoc multiple comparison test. *p*-Values are indicated in the figure legend. Data information: n = number of biological replicates; * *p* ≤ 0.05, ** *p* ≤ 0.01, *** *p* ≤ 0.001, **** *p* ≤ 0.0001.

## 3. Results

### 3.1. Description of Experimental Results

#### 3.1.1. USP8 Down-Regulation Induces Autophagy in Flies

To dissect the molecular pathway underlying USP8 down-regulation, we performed a mass-spectrometry-based analysis of USP8-deficient flies to identify the repertoire of USP8 substrates. To this aim, protein lysates extracted from 200 WT and USP8 knock-down flies were subjected to immunoaffinity isolation and mass spectrometry (MS) analysis to enrich and identify K-GG peptides from digested protein lysates [27]. This analysis identified 1149 modified peptides in WT and 940 in USP8 knockdown flies with a significant (>1.0) VML score (Appendix A). Among these, we identified 254 peptides that are ubiquitinated in USP8 down-regulating flies only (Appendix A) and 347 that are only ubiquitinated in WT flies (Appendix A). In the subset that was found to be ubiquitinated only in USP8 down-regulating flies, gene ontology analysis identified enrichment in ubiquitinated proteins that belong to signaling pathways regulating tissue differentiation and development (Hedgehog, dorsoventral axis formation and FoxO signaling pathways) and components of the mitophagy pathways (Appendix A). Interestingly, among the identified ubiquitinated fragments that are unique to USP8-deficient flies, the MS analysis identified several mitochondrial proteins (Appendix A), including transmembrane GTPase Marf (fly orthologue of Mitofusin), targets of mitophagic protein Parkin [28] and MUL1 [5] and Porin/VDAC, which was previously identified as a Parkin target [29]. This analysis also identified proteins that are involved in vesicular trafficking, consistent with a role of USP8 in endosomal trafficking and transport (Ras85D and Protein Star, both involved in EGFR signaling, and Flo1), and autophagy regulators (for example Rab3 GTPase activating protein, scny, Gyf). Most of the identified proteins are transcriptional factors or regulators of chromosome organization and segregation, consistent with the activation of key transcriptional pathways highlighted by gene ontology. Considering the correlation between USP8 and autophagy, which was reported in previous publications, and the identification in our MS analysis of autophagic and mitophagic factors, we wanted to evaluate autophagy levels when USP8 activity is altered. To achieve this, we used fly genetics to generate several fly lines expressing the fluorescent autophagic flux probe GFP-mCherry-Atg8a (WT controls, USP8 RNAi, and USP8^−/+^) under the control of the pan-neuronal driver nSybGAL4. The probe allows discriminating between autophagosome (green + red fluorescence) and autolysosome (red fluorescence) by confocal fluorescent microscopy [30]. The acidic environment of the autolysosome quenches the GFP signal and promotes the fluorescence switch that occurs upon autophagosome–lysosome fusion (Figure 1A). We dissected larval brains and counted the number of autolysosomes and autophagosomes per cell, based on the fluorescence signal. In the larval brain of WT flies, genetic down-regulation of USP8 (both RNAi and USP8^−/+^) (Appendix A) resulted in an increased number of autolysosomes, whereas the number of autophagosomes did not change (Figure 1B–D). This result can be interpreted as an overall increase in the autophagic flux upon USP8 knockdown in the larval brain of WT flies. Transmission electron microscopy (TEM) analysis from thoracic muscle of WT and USP8-down-regulating flies revealed a higher number of autophagic vesicles (autophagosomes and autolysosomes) in USP8-down-regulating conditions (Figure 1E,F), fully supporting the autophagic effect of USP8 down-regulation.

Next, we wanted to investigate the autophagic effect of USP8 down-regulation in a model of neurodegeneration in which loss of organelle homeostasis is implicated: the Parkin KO flies. At the systemic level, Parkin KO (Park^25^) flies develop disorganized muscle fibers with irregular arrangement of myofibrils, locomotor dysfunction, and reduced lifespan. At the cellular level, these flies are characterized by specific loss of dopaminergic neurons (DA) in the PPL1 cluster and widespread mitochondrial abnormalities, which correlate with impaired mitochondrial respiration and function. Importantly, USP8 down-regulation enhanced the autophagic flux in the larval brain of Parkin KO flies (Figure 1G–I).

In conclusion, USP8 down-regulation enhances the autophagic flux in WT flies. In Parkin KO flies, USP8 down-regulation also increases autophagy levels, supporting the hypothesis of a protective effect of USP8 down-regulation that depends on autophagy.

#### 3.1.2. USP8 Down-Regulation Induces Mitophagy in Flies

Our results indicate that autophagy is enhanced in USP8-deficient flies, what about mitophagy? We measured the mitophagic flux in the Drosophila brain by taking advantage of newly generated lines expressing the mitophagic fluorescent reporter probe mito-QC [25]. All lines were characterized by the presence of the *UAS-mitoQC* reporter in the second chromosome and the *nSyb-GAL4* driver in the third chromosome to allow mitophagy evaluation by the mito-QC approach in larval neurons. Similar to the autophagic flux reporter, mito-QC is a tandem mCherry–GFP probe targeted to the outer mitochondrial membrane (OMM) that labels mitochondria red-green. When the organelle is delivered to the lysosomes, the acidic environment of the lysosome quenches the GFP fluorescence, whereas the mCherry signal remains stable. Therefore, mCherry (red)-only puncta can be interpreted as “acidic” mitochondria (mitochondria that are delivered to the lysosomes, i.e., mitolysosomes), and their number and size can be used as read out for on-going mitochondrial degradation (Figure 2A). We determined basal mitophagy levels analyzing neurons in the ventral nerve chord (VNC) of third instar stage larvae. In WT flies, we observed on average four to five mitolysosomes per cell, whereas in the brains of USP8-down-regulating flies, data analysis showed a significant increase in the number of mitolysosomes, indicative of enhanced mitophagy in this condition (Figure 2B,C). Importantly, this effect was also induced in a Parkin KO background, indicating that the mitophagic effect of USP8 down-regulation is Parkin independent (Figure 2D,E).

Because mitochondria undergo a tight remodeling of their shape and ultrastructure when mitophagy is induced [31], we evaluated whether USP8 down-regulation affected mitochondrial architecture. To this aim, we dissected the thoracic muscle of WT and USP8 down-regulating flies that contained a large number of mitochondria and processed the samples for TEM analysis. In the fly muscle, mitochondria appear as electron dense structures placed in between the muscle myofibrils. To determine the shape of individual mitochondria, we measured the mitochondrial aspect ratio (AR). To this aim, an ellipse is fitted to the mitochondrion and the major (longitudinal length) and minor axis (equatorial length) of the ellipse is used to calculate AR = l_major_/l_minor_. Because the shape of most mitochondria resembles an ellipsoid shape, calculation of the AR yields a reliable approximation of the elongation of a given mitochondrion. Intuitively, the smaller the AR is the more fragmented the mitochondrial network will be. We evaluated the AR of these structures, and found a significant decrease in the AR of mitochondria in USP8-down-regulating flies, which indicates increased mitochondrial fission (Figure 2F,G). In this analysis, we used Mfn/Marf RNAi flies as reference for the evaluation of the fragmented phenotype. These results fully support what we previously observed in USP8-down-regulating S2R+ cells that displayed fragmented mitochondria [20]. Mitochondrial fission is known to promote uncoupled respiration as a means to reduce oxidative stress, which, when elevated, triggers mitochondrial quality control mechanisms to remove damaged mitochondrial components. Thus, this morphological change likely facilitates quality control, whereas it does not seem to affect mitochondrial functionality, in that our previous study showed that mitochondrial respiration and Complex I activity are not affected in USP8-down-regulating conditions [20].

In summary, USP8 down-regulation correlates with enhanced levels of basal mitophagy. This effect is Parkin independent. Importantly, USP8-deficient flies did not show any particular detrimental phenotypes and mitochondrial functionality remained intact [20].

#### 3.1.3. USP8 Down-Regulation Enhances Basal Mitophagy in S2R+ cells

We next wanted to dissect in more detail the molecular pathway underlying the mitophagic effect of USP8 down-regulation. In order to do that, we moved to an in vitro cell model and treated S2R+ fly cells with Ctrl or USP8 dsRNA to specifically knockdown USP8. Upon efficient USP8 down-regulation in fly cells (Appendix A), we measured mitophagy progression and occurrence by looking at three essential steps that characterize the process of mitochondrial degradation via autophagy: (i) interaction between mitochondria and the autophagic factor LC3; (ii) delivery of the organelle to hydrolase-containing lysosomes and (iii) actual degradation of mitochondrial resident proteins. To evaluate step one, we transfected S2R+ cells with the phagophore and autophagosome marker RFP-LC3 for two days and loaded cells with mitochondrial fluorescent probe MitoTracker Green^®^. We performed live imaging confocal microscopy on these samples. In USP8-down-regulating conditions, we observed increased levels of LC3-positive phagophores co-localizing with mitochondria (Figure 3A,B). To evaluate step two, we took advantage of the mt-Keima probe, a pH sensitive fluorescent probe targeted to the mitochondrial matrix that has different excitation spectra at neutral (405/615 nm) and acidic pH (561/615 nm) (Figure 3C). We transfected S2R+ fly cells with mt-Keima before treating cells with Ctrl and USP8 dsRNA. USP8 downregulated cells showed a clear shift in spectra, with a significant increase in the average signal at 561 nm (i.e., acidic pH) (Appendix A) resulting in an increase in the 561/405 ratio (Figure 3D). This result indicates that acidification of the mitochondrial matrix is occurring in this condition, which means increased mitochondrial material that has been delivered to the acidic environment of the lysosome. We next measured the actual degradation of the organelle by looking at protein levels of mitochondrial-resident proteins: TOM20 and VDAC for OMM, Cyclophilin D (CyPD) for mitochondrial matrix and ATPase/Complex V (CV) for inner mitochondrial membrane (IMM) (Figure 3E). The assumption here is that if mitochondria are degraded and this is not compensated by productive mitochondrial biogenesis, mitochondrial loss should be reflected by a decrease in the levels of these proteins. In USP8-down-regulating conditions, we found that TOM20, VDAC, CyPD and CV were only slightly decreased (Figure 3F,G). Because the reduction in mitochondrial content is quite modest, and USP8 downregulating cells do not display any defects in mitochondrial respiration [20], it is possible that mitochondrial degradation induced by USP8 down-regulation is compensated for by productive mitochondrial biogenesis. In this scenario, the mitophagic effect of USP8 down-regulation might not necessarily translate into an obvious decrease in mitochondrial mass.

As previously discussed, one of the best-characterized ubiquitin-dependent mitophagy pathways depends on the activation of E3-ubiquitin ligase Parkin [32]. However, our in vivo results indicate that the proteostatic (autophagic/mitophagic) effect of USP8 down-regulation occurs under basal conditions and in Parkin KO background. Our previous studies showed that USP8-deficient cells do not display any defects in mitochondrial respiration, nor loss of mitochondrial membrane potential to trigger PINK1/Parkin activation [20]. Thus, it seems unlikely that the mitophagic effect of USP8 down-regulation correlates with Parkin activation. Nevertheless, to exclude this possibility, we evaluated whether Parkin thermal stability and Parkin mitochondrial recruitment, two key elements underlying Parkin activation, are affected upon USP8 down-regulation. We found that USP8 down-regulation correlates with an increased thermal stability for Parkin, indicating that Parkin is actually more stable in this condition (Appendix A). Additionally, in USP8 downregulating cells, Parkin did not translocate to mitochondria, even when cells were challenged with CCCP to trigger Parkin translocation (Appendix A). In this condition, basal mitophagy is enhanced (Figure 3D), whereas stress-induced mitophagy triggered by CCCP (10 μM/20 h) is reduced (Appendix A). The fact that stress-induced mitophagy is inhibited in USP8 down-regulating conditions is in agreement with a previous study showing that USP8 is required for Parkin translocation and stress-induced mitophagy [17].

These results support the hypothesis that basal mitophagy is induced by USP8 down-regulation and that the mitophagic effect of USP8 down-regulation under basal conditions is Parkin independent.

#### 3.1.4. USP8 Inhibition Induces Mitophagy in Neurons of Human Origin

Extending these findings to mammals, we next evaluated the effect of USP8 inhibition in primary neurons of human origin. To this end, we used H9 human embryonic stem cells (hESCs) to obtain neurons by forcing the expression of transcription factor Neurogenin-2 (Ngn2) under the control of a TetO promoter induced by doxycycline [22]. Ngn2 regulates the commitment of neural progenitors to neuronal fate during development [33] and induces early postnatal astroglia into neurons [34]. It is known that overexpression of Ngn2 and Sox11 (another transcription factor involved in neuronal induction during embryonic development) promote the differentiation of primary fibroblasts into cholinergic neurons, whereas they inhibits GABAergic neuronal differentiation [35]. Ngn2 expression in hESCs produces an excitatory layer2/3 cortical neuron that exhibits AMPA-receptor-dependent spontaneous synaptic activity and a relatively smaller NMDA-receptor-mediated synaptic current [22]. These iNeurons express glutamatergic synaptic proteins such as vesicular glutamate transporter 1 (vGLUT1), postsynaptic density-95 (PSD95) and synapsin1 (SYN1) and excitatory synaptic function when in co-culture with mouse glial cells [22]. The yield of neuronal conversion is nearly 100% and, most importantly, this protocol allows generating primary neurons with reproducible properties in only two weeks. After 4 days of differentiation, the cells start to develop a clear neuronal network (Appendix A), and at the end of the differentiation process (14 days), iNeurons exhibit the expression of the typical neuronal markers MAP2 and βIII-tubulin and lose pluripotency markers OCT4 and sox2 (Appendix A). Quantitative RT-PCR analyses revealed that iNeurons expressed ~30- to ~100-fold increased levels of endogenous Ngn2 as well as of three neuronal markers, NeuN, MAP2 and Tuj1, compared with H9 ESCs (Appendix A). Immunoblotting experiments confirmed that stem cell marker OCT3/4 is only present until day 2 of differentiation, whereas the expression of neuronal marker βIII-Tubulin gradually increased until day 14 upon induction (Appendix A). Our representative electron microscopy (EM) images of iNeurons after 14 days of differentiation show neuronal cells with distinguishable neuronal soma, axon hillock and axonal and dendritic projections (Appendix A). Released neurotransmitter molecules are visible at synaptic clefts (Appendix A) and detectable levels of NMDA-R are expressed, which we assessed by Western blotting analysis (Appendix A). Thus, these cells fully develop as neurons and seem to make functional synapsis. We generated hESCs expressing the fluorescent mitophagic probe mtx-QC^XL^ [36] and differentiated them into iNeurons. Mtx-QC^XL^ is a matrix-targeted mCherry–GFP protein that allows monitoring ongoing mitophagy by fluorescent microscopy, in that delivery of mtx-QC^XL^ to lysosomes leads to selective accumulation of mCherry-positive fluorescence as a result of GFP quenching (Figure 4A). We generated hESCs line stably expressing the mtx-QC^XL^, which we differentiated into iNeurons upon treatment with doxycycline (Figure 4B). We treated iNeurons with the specific UPS8 inhibitor DUBs-IN-2 [21] to mimic the catalytic inactivation of USP8. In these neurons of human origin, we found that USP8 pharmacological inhibition by DUBs-IN-2 (0.5–1 μM/24–48 h) enhances basal mitophagy, an effect that was comparable to 0.5 μM antimycin/oligomycin (AO) treatment, which we used as positive control for mitophagy induction (Figure 4C). Interestingly, iNeurons seem to display significant levels of basal mitophagy (Figure 4C).

In conclusion, these data support the hypothesis of a mitophagic effect of USP8 down-regulation that we observed in vivo in flies and in several cell lines, including neurons of human origin.

**Figure 3 cells-12-01143-f003:**
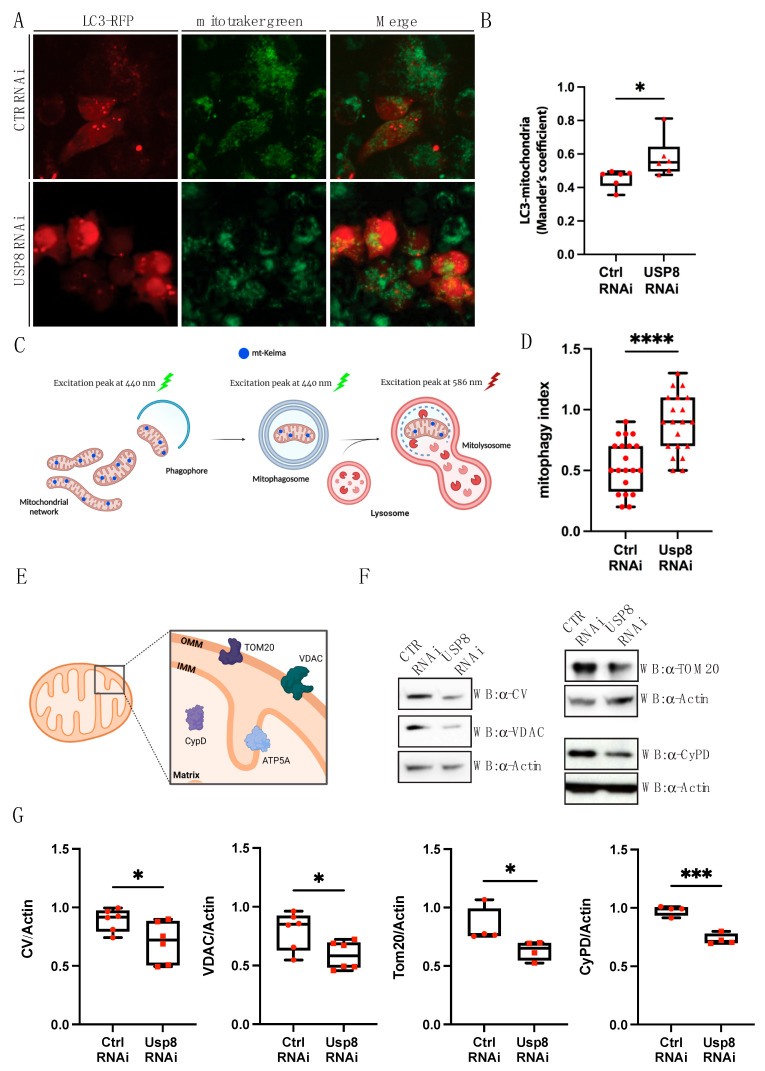
(**A**) Representative confocal images of cells transfected with LC3–RFP and loaded with MitoTracker Green^®^ to study mitochondria–autophagosome co-localization. Cells were treated with the indicated dsRNA (CTR and USP8), and after 24 h they were transfected with LC3–RFP for two days. After transfection, cells were loaded with MitoTracker Green^®^ and imaged with a spinning disk confocal microscope, using a UPlanSApo 60×/1.35 NA objective (iMIC Adromeda, TILL Photonics). (**B**) Quantification of (**A**). Autophagosome–mitochondria interaction was evaluated by measuring the percentage of LC3 that co-localizes with mitochondria by Mander’s coefficient of co-localization using ImageJ JACoP (“Just Another Colocalization Plug in”), following 3D volume rendered reconstruction of 40 *z*-axis images, separated by 0.2 μm. Chart shows mean ± SEM of *n* = 3 replicates. Statistical significance was determined by unpaired *t*-test; * = *p* < 0.05. (**C**) Graphical representation of the mt-Keima construct. A fluorescent Keima protein is targeted to the mitochondrial matrix. Under neutral conditions, mt-Keima has an excitation peak at 440 nm, but in an acidic environment, such as that of lysosomes, the excitation peak shifts to 586 nm. The ratio 561/458 nm allows quantification of on-going mitochondrial degradation (adapted from [37]). Created with BioRender.com. (**D**) Mitophagy levels assessed by flowcytometry using the mt-Keima probe. Data are represented as fold change compared with control conditions. Chart shows mean ± SEM of *n* = 11 replicates. Statistical significance was determined by unpaired *t*-test. (**E**) Graphical representation of mitochondrial proteins and their localization. Created with BioRender.com. (**F**) Representative Western blotting analysis of the indicated proteins in control and USP8 RNAi cells. (**G**) Quantification of (**F**). VDAC (number of replicates, *n* = 6) and TOM20 (*n* = 4) were used as representative of OMM-resident proteins; CV (*n* = 6) was used as representative of IMM-resident proteins; CyPD (*n* = 4) was used as representative of for matrix-resident proteins. Actin was used as loading control. Data are represented as fold change compared with control conditions. Charts show mean ± SEM. Statistical significance was determined by unpaired *t*-test; * = *p* < 0.05; *** = *p* < 0.001; **** = *p* < 0.0001.

**Figure 4 cells-12-01143-f004:**
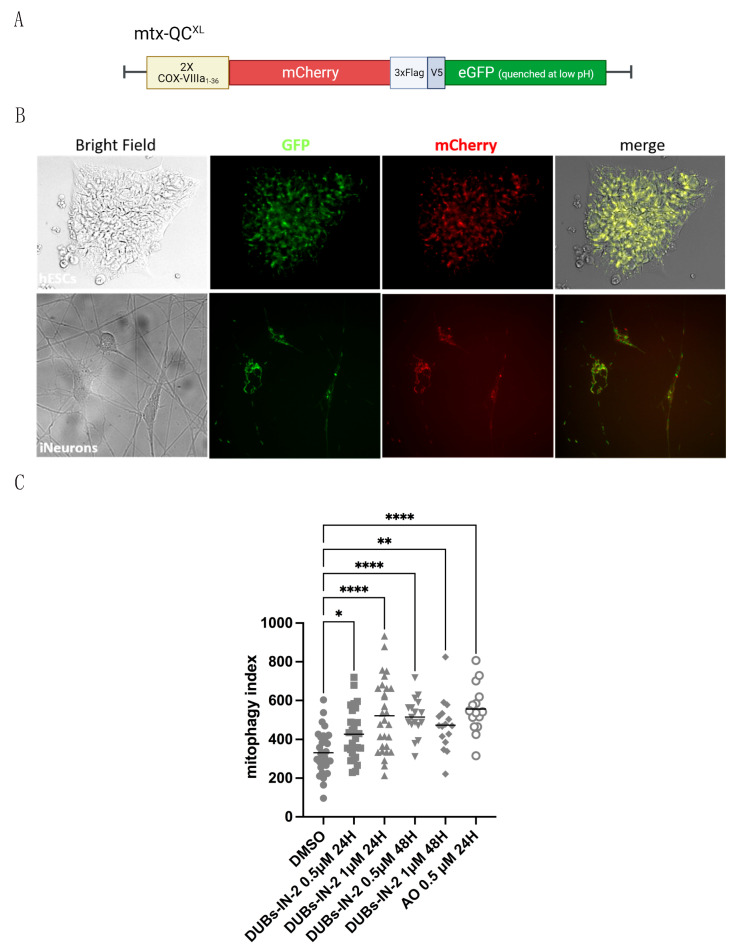
(**A**) Graphical representation of the mito-QC construct (mtx-QC**^XL^**). A tandem mCherry–GFP tag is targeted to the IMM via mitochondrial targeting sequence of the Cytochrome *c*
**oxidase subunit 8A (COX VIIIA)**. Under normal conditions, mitochondria display both mCherry and GFP fluorescence. The GFP signal is quenched upon mitochondria fusion to lysosomes. In the acidic environment of the lysosomes the number of mCherry-only foci (red dots) allows quantification of on-going mitochondrial degradation. Created with BioRender.com (**B**) hESCs were transfected with Mtx-QC^XL^, and several clones stably expressing the probe were generated upon antibiotic selection. Mtx-QC^XL^ is a matrix-targeted mCherry–GFP protein that stains mitochondria in yellow (modified from [36]). Delivery of mtx-QC^XL^ to lysosomes leads to selective accumulation of mCherry-positive fluorescence as a result of GFP quenching, thus allowing monitoring of on-going mitophagy by fluorescent microscopy. (**C**) Mitophagy analysis in mtx-QC^XL^-expressing iNeurons upon treatment with specific UPS8 inhibitor DUBs-IN-2 to mimic catalytic inactivation of USP8. In these neurons of human origin, USP8 pharmacological inhibition by DUBs-IN-2 (0.5–1 μM/24–48 h) enhances basal mitophagy. Antimycin/oligomycin (AO) treatment was used as positive control for mitophagy induction. * = *p* < 0.05; ** = *p* < 0.01; **** = *p* < 0.0001.

## 4. Discussion

Loss of proteostasis is well documented during aging, in part as a consequence of the progressive physiological decline in the proteolytic activity of two major degradative systems: the ubiquitin–proteasome and the lysosome–autophagy systems. Whereas a physiological decline in proteostasis is expected in aged individuals, in age-associated neurodegenerative conditions this drop seems to be pathologically exacerbated [1]. What are the reason for this? In the quest for the potential regulators of proteostasis that might be affected in neurodegenerative conditions, deubiquitinating enzymes (DUBs) are interesting candidates for their fine-tuning activity on the ubiquitination statuses of proteins. DUBs are proteases that counteract ubiquitination by cleaving ubiquitin moieties from proteins. Given that one of the main functions of ubiquitination is to promote protein degradation, as well as bridging the ubiquitin proteasome system (UPS) to autophagy and mitophagy, specific DUB down-regulation can presumably enhance protein degradation, autophagy and basal mitophagy and be beneficial in neurodegenerative diseases in which accumulation of misfolded proteins and aberrant mitochondria is implicated. One interesting DUB in this context is the ubiquitin-specific protease USP8, as its knockdown or pharmacological inhibition is protective in different models of neurodegeneration. USP8 knockdown decreases Amyloid β (Aβ) production in an in vitro model of AD, presumably by promoting lysosome-dependent degradation of β-secretase, the enzyme involved in amyloid precursor protein (APP) processing [18]. USP8 down-regulation also protects from α-synuclein-induced toxicity in an α-synuclein fly model of PD [19], and its down-regulation or pharmacological inhibition ameliorates the phenotype of PINK1 and Parkin KO flies [20]. Interestingly, USP8 is highly expressed in the brain and specifically in dopaminergic neurons. Moreover, its levels seem to be inversely correlated with the extent of Lewy Body (LB) ubiquitination in post-mortem brains of PD patients [19]. This evidence indicates a protective effect of USP8 down-regulation, which might depend on its proteostatic activity or other activities correlated to USP8 pleiotropic functions.

Whereas the functions of USP8 have been extensively explored in the context of EGFR endocytosis in different cell types and in the regulation of stem cell proliferation and self-renewal in stem cells, the consequences of USP8 manipulation in post-mitotic, long-lived cells such as neurons is poorly defined.

Thus, in this work we wanted to explore the possibility of a proteostatic effect of USP8 down-regulation in neurons.

We started by taking an unbiased approach to determine the repertoire of USP8 substrates and identify signaling pathways that are specifically altered upon USP8 down-regulation. To this aim, we generated fly lines stably down-regulating USP8 and performed a mass spectrometry (MS)-based analysis from protein lysates extracted from WT and USP8-down-regulating flies and subjected them to immunoaffinity isolation to enrich K-GG peptides [38]. The method is based on the identification of di-glycine (GG) ubiquitin remnants that are left on lysine (K) residues after trypsinization and allows identifying ubiquitinated fragments, which should be specifically enriched in USP8-down-regulating conditions. Flies are an ideal model to do so because, as opposed to mice in which transcription and/or translation of the intact allele does compensates for the loss of one gene copy, USP8 down-regulation or hemizygosity in the fly exhibits reduced protein levels.

Gene ontology analysis on the ubiquitinated proteins identified signaling pathways regulating tissue differentiation (dorsoventral axis formation, Hegdehog signaling pathway, Foxo signaling pathway). Surprisingly, mitophagy was among the ten highest scoring KEGG pathways that came out from this analysis. Of particular relevance for us, among the hits that scored a significant VML index were Marf, fly ortologue of mitochondrial pro-fusion protein Mitofusin and Porin/VDAC. Both proteins are key regulators of mitochondrial quality control, in that Parkin-dependent ubiquitination of Marf/Mfn or VDAC signals selected mitochondria for degradation. Based on these results, the next step was to investigate the potential autophagic and mitophagic effect of USP8 down-regulation by taking advantage of fluorescent probes that allow measuring the autophagic and mitophagic flux in the drosophila brain in combination with fly genetics. These approaches allowed identifying a mitophagic effect of USP8 down-regulation, which was clearly detectable in vivo in the fly brain and also in neurons of human origin.

Interestingly, USP8 down-regulation promoted basal mitophagy in a Parkin-independent fashion (Figure 2D,E), whereas it inhibited Parkin mitochondrial translocation and mitophagy under stress condition (Appendix A). The latter was in agreement with a previous study showing that USP8-dependent deubiquitination of Parkin is required for Parkin recruitment and activation following CCCP-induced mitochondrial stress [17]. Thus, USP8 seems to play a role in both Parkin-independent (basal) mitophagy and stress-induced mitophagy triggered by mitochondrial depolarization. The molecular mechanism of mitophagy induction under basal conditions is unknown. Nevertheless, our MS data suggests that Marf/Mfn is a possible target for USP8 (Appendix A), thus providing a mechanistic insight into the molecular mechanism of mitophagy induction, which might be via a USP8 effect on the ubiquitination levels of Marf/Mfn. Marf/Mfn is a mitochondrial fusion protein with pleiotropic functions [39,40] that is also a target of Parkin-dependent ubiquitination upon stress-induced mitophagy. Marf/Mfn is also target of E3 ubiquitin ligases other than Parkin, for example MUL1 [5]. It is possible that MUL1 operates as the E3 ubiquitin ligase regulating the ubiquitination of Marf/Mfn and mitophagy, in opposition to USP8. Further studies are required to clearly dissect the molecular mechanism of mitophagy induction triggered by USP8 down-regulation and its physiological importance.

Of particular relevance for a potential therapeutic application of USP8 down-regulation, potent and highly specific inhibitors of USP8 are available, which were generated based on the USP8 crystal structure. The best inhibitors at present were developed as derivatives of 9-oxo-9H-indeno[1,2-b]pyrazine-2,3-dicarbonitrile [21]. Detailed pharmacokinetic data and dosing regimes are available. These compounds (DUBs-IN-2 and DUBs-IN-1) have an IC_50_ value in the range of 200 nM and are highly specific for USP8 (e.g., IC_50_ value of >100 μM for Usp7). Both inhibitors kill HCT116 colon cancer cells and PC-3 prostate cancer cells, and DUBs-IN-2 has been used to diminish tumorigenesis in breast cancer [7] and in corticotroph tumor cells [8]. DUBs-IN-2 seems to be well tolerated in vivo in rodents, and it has been safely used to treat gastric cancer in mice [41]. Thus, important prerequisites for compound optimization and drug development exist for USP8 and can be readily exploited in aged-associated neurodegenerative disease models.

In summary, in this work we show that we can enhance autophagy and mitophagy by down-regulating USP8, a DUB that is upregulated in age-related neurodegenerative conditions [19,42]. Many studies have shown that promoting autophagy increases lifespan and rescues the pathological phenotype of animal models of neurodegeneration, supporting the hypothesis of a protective effect of enhanced proteostasis to prevent neuronal loss [1]. Among the proteostatic mechanisms that might have therapeutic implications for the treatment of neurodegenerative conditions, mitophagy plays a crucial role. Indeed, one proposed underlying mechanism of neurodegeneration includes alterations in mitochondrial function and increased oxidative stress that can affect the proteostatic capacity of the cell [43]. In this scenario, approaches that enhance mitochondrial quality control, such as mitophagy, might be beneficial to degrade dysfunctional mitochondria as sources of potentially toxic compounds.

Our work provides a mechanistic explanation for the protective effect of USP8 down-regulation that is via enhancement of mitophagy and lays the basis for further development of studies targeting DUBs (USP8 in particular) in neurodegenerative conditions.

## Figures and Tables

**Figure 1 cells-12-01143-f001:**
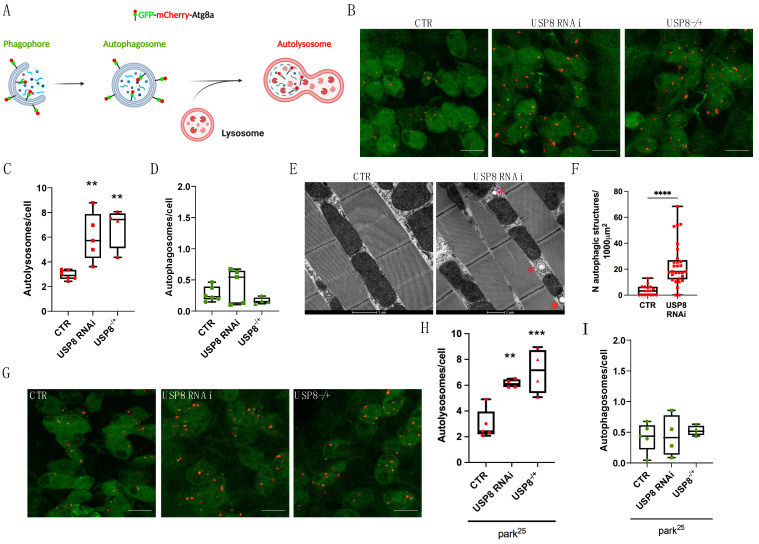
(**A**) Graphical representation of the GFP–mCherry–Atg8a construct. A tandem mCherry–GFP tag is fused to Atg8a. In conditions of neutral pH such as the cytosol, autophagosomes display both mCherry and GFP fluorescence. The GFP signal is quenched upon autophagosomes fusion to lysosomes. In the acidic environment of the lysosomes, the number of mCherry-only foci (red dots) allows quantification of on-going autophagy. Created with BioRender.com. (**B**) Confocal microscopy analysis of larval VNC neurons expressing GFP–mCherry–Atg8a. mCherry-only puncta represent autolysosomes under basal condition (CTR) or upon USP8 down-regulation (USP8 RNAi and USP8^−/+^). (**C**) Quantification of autolysosomes (red-only dots) per cell in the three different conditions. Statistical significance determined by one-way ANOVA with Dunnett’s post-test correction; ** = *p* < 0.01. (**D**) Quantification of autophagosomes per cell in the three different conditions. Statistical significance determined by Kruskal–Wallis test with Dunn’s multiple comparison. (**E**) Representative electron microscopy images of flight muscle mitochondria of the indicated genotypes; * indicate autophagosome stractures. (**F**) Quantification of (**E**). Box plot represents quantification of autophagic vesicles formation in the flight muscle of the indicated genotypes. Statistical significance determined by Student’s *t* test. **** = *p* < 0.0001. (**G**) Confocal microscopy analysis of larval VNC neurons expressing GFP–mCherry–Atg8a in park^25^ flies. (**H**) Quantification of autolysosomes (red-only dots) per cell in the three different conditions. Statistical significance determined by one-way ANOVA with Dunnett’s post-test correction; ** = *p* < 0.01; *** = *p* < 0.001. (**I**) Quantification of autophagosomes per cell in the three different conditions. Statistical significance determined by Kruskal–Wallis test with Dunn’s multiple comparison. Unless differently indicated, scale bars = 10 µm. Genotypes analysed (confocal microscopy): UAS GFP-mCherry-Atg8a/+; nSybGAL4/+ (CTR), UAS GFP-mCherry-Atg8a/+; nSybGAL4/UAS USP8 RNAi (USP8 RNAi) and UAS GFP-mCherry-Atg8a/+; nSybGAL4/USP8 KO (USP8^−/+^). Genotypes analysed (TEM): Act5cGAL4/+ (CTR), Act5cGAL4/+; UAS USP8 RNAi/+ (USP8 RNAi).

**Figure 2 cells-12-01143-f002:**
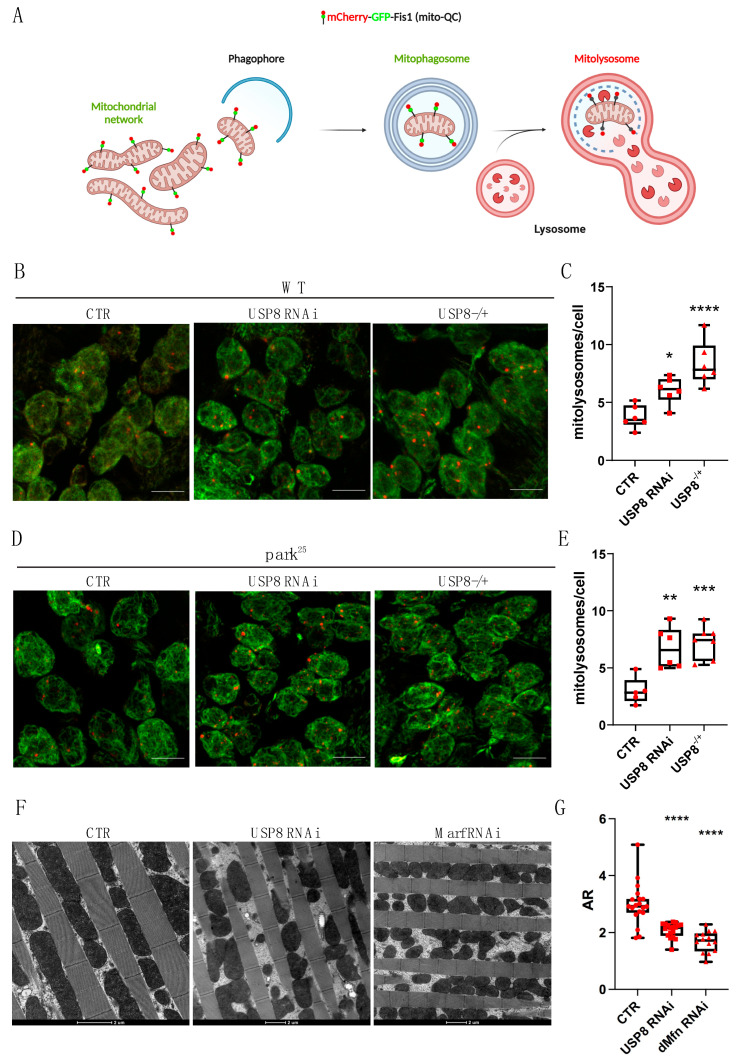
(**A**) Graphical representation of the mito-QC construct. A tandem mCherry–GFP tag is targeted to the OMM. Under normal conditions, mitochondria display both mCherry and GFP fluorescence. The GFP signal is quenched upon mitochondria fusion to lysosomes. In the acidic environment of the lysosomes the number of mCherry-only foci (red dots) allows quantification of on-going mitochondrial degradation. Created with BioRender.com. (**B**) Confocal microscopy analysis of larval VNC neurons expressing mito-QC. mCherry-only puncta represent mitolysosomes under basal conditions (CTR) or upon USP8 down-regulation (USP8 RNAi and USP8^−/+^). (**C**) Quantification of mitolysosomes per cell in the three different conditions. Statistical significance determined by one-way ANOVA with Dunnett’s post-test correction; * = *p* < 0.05; **** = *p* < 0.0001. (**D**) Confocal microscopy analysis of larval VNC neurons expressing mito-QC in park^25^ flies. (**E**) Quantification of mitolysosomes per cell in the three different conditions. Statistical significance determined by one-way ANOVA with Dunnett’s post-test correction; ** = *p* < 0.01; *** = *p* < 0.001. (**F**) Representative electron microscopy images of flight muscle mitochondria of the indicated genotypes. (**G**) Quantification of (**F**). Box plots represent quantification of mitochondria aspect ratio of the indicated genotypes. Statistical significance determined by one-way ANOVA with Dunnett’s post-test correction. **** = *p* < 0.0001. Unless differently stated, scale bars = 10 µm. Genotypes analysed (confocal microscopy): UAS mito-QC/+; nSybGAL4/+ (CTR), UAS mito-QC/+; nSybGAL4/UAS USP8 RNAi (USP8 RNAi), UAS mito-QC/+; nSybGAL4/USP8 KO (USP8^−/+^), UAS mito-QC/+; nSybgal4, park^25^/park^25^ (park^25^ CTR), UAS mito-QC/+; nSybGAL4, park^25^/UAS USP8 RNAi, park^25^ (park^25^ USP8 RNAi), UAS mito-QC/+; nSybGAL4, park^25^/USP8 KO, park^25^ (park^25^ USP8^−/+^). Genotypes analyzed (TEM): Act5cGAL4/+ (CTR), Act5cGAL4/+; UAS USP8 RNAi/+ (USP8 RNAi); Act5cGAL4/UAS Marf RNAi (Marf RNAi).

## Data Availability

Not applicable.

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
