# Peer review of "USP8 Down-Regulation Promotes Parkin-Independent Mitophagy in the Drosophila Brain and in Human Neurons"

_cells, 2023, doi:10.3390/cells12081143_

Round 1

Reviewer 1 Report

USP8 has been shown to be involved in Parkin-mediated mitophagy (2014, EMBO J) and hence, it is a little surprising that the authors report here that USP8 is required for Parkin-independent mitophagy. Unfortunately, their data are very thin and not convincing. Mainly, they just described the enhanced mitophagy phenotype caused by USP8 RNAi in flies, S2R cells and iNeuron cells without any further mechanistic studies. How does USP8 RNAi promotes mitophagy? Is the DUB activity required? Does it affect BNIP3/NIX protein level, two proteins known to be involved in Parkin-independent mitophagy?

The figures can be consolidated into 3 as Figure 1 and 2 are kind of duplicative and Figure 4 can be moved into supplementary. Introduction lacks enough background information and some of the information in the Discussion should be moved into Introduction. For instance, they only mentioned USP8 is embryonic lethal in the discussion. It should be described in the Introduction or early in the Results to explain why USP8 +/- flies not USP8-/- flies were used. This also raised an important question about their Mass Spec data conducted in the USP8+/- flies. Does it mean USP8 is haploinsufficient?  Do USP8+/- flies exhibit any developmental defects? Actually in their 2019 paper, they reported that USP8 RNAi or USP8+/- or USP8 specific inhibitor can rescue PINK1/Parkin KO phenotype. These data should be described more in Introduction to give readers a better context. 

Is it possible that USP8 plays a role in both Parkin-dependent and -independent mitophagy? In Figure 3H, the authors do show that USP8 RNAi does block Parkin translocation, which confirmed Durcan et al (2014) data. The authors could have expanded this experiment by treat S2R+ cells with CCCP for 24 hrs to see if USP8 RNAi inhibits CCCP-induced mitophagy while enhances basal Parkin-independent mitophagy.

All the RNAi experiment should include a WB probed with anti-USP8 and rescue data (at least for S2R+ cells).

KEGG pathway: mitophagy p-value is only 0.07635 and adjusted p-value is 0.2405. Is this significant?

Fig.1 C-D. The number of autophagosomes/cell is significantly less than the number of autolysosomes/cell. Does it mean basal mitophagy is pretty robust already? They should check GFP-mCherry-ATG8a signal in EPG5 or other mutants that known to block autophagosome formation. This is very important since park25 mutant fly has similar basal level of autolysosome or mitolysosome formation.

Fig.3D. WB is not convincing. A better WB should be provided. anti-USP8 should be probed to verify the RNAi level. Bafilomycin treatment should be done to show the degradation is dependent on autophagy. 

Fig. 3B. FACS plot should be provided. 

Both Introduction and Discussion need to be improved. They lack clear flow and transition between each paragraph is often not smooth. It's hard to follow their messages.

Reviewer 2 Report

The manuscript by Mauri et al. describes the physiological role of the ubiquitin-specific protease USP8 and the effects of its downregulation on mitophagy, by analysing fly models and human cells. The authors speculate about inhibition of USP8 as therapeutic strategy for neurodegenerative diseases, mainly for Parkinson Disease (PD).

The paper is interesting and reported a possibly relevant new pathway involved in mitophagy, that could have a translational consequence for monogenic, and perhaps idiopathic, PD.

1)     However, I have some objections mainly concerning the strategies used for USP8 downregulation.

I don’t like the expression “genetic inhibition of USP8”. I think reduction or downregulation is more appropriate: inhibition seems related to the function rather than the amount, but the models (RNAi and heterozygotes) have a quantitative defect. Even I would not use “USP8 deficient mouse” for partial reduction.

The level of USP8 downregulation obtained by RNAi has to be shown, both for the transcript and possibly also for the protein. The authors just wrote “upon efficient USP8 downregulation”

There is inconsistency of the types of cells used for different experiments, i.e. neurons in the ventral nerve chord (VNC) and flight muscle in figs. 1-2. Please, provide a brief explanation of why.

Finally, the authors used pharmacological inhibition only on neurons. Why?

2)     PINK1

As reported in the introduction (l. 65-68), Parkin is not indispensable for mitophagy, but PINK1 seems to have a more crucial role. It would be interesting to have some data on PINK1 levels in the USP8 models.

3)      MS analysis:

in addition to focusing on proteins ubiquitinated in USP8 mutant flies only, a brief description of the protein ubiquitinated on WT flies only is interesting (as indication of the physiological role of USP8).

4)     PARK2 mutant flies

Is it expected that there is no difference between wt and park2 mutant flies for authophagosomes (fig.1 panel C vs E; fig.2 panel C vs E)

5)     Mitochondrial degradation

The reduction in mitochondrial protein levels (Figure 3D-E) is quite modest, and the densitometry quantification is not highly sensitive. I cast doubt it has a deleterious effect. Accordingly, the authors reported that previous studies “showed that USP8 deficient cells do not display any defects in mitochondrial respiration” (l 475).

Other functional assays are needed to confirm that these models have a relevant decrease in mitochondrial mass: e.g. reduction in complex V subunits would lead to ATP deficiency. Assessment of mtDNA amount is another way to support increased mitophagy, without using immunoblots.

6)     Mitochondrial network

The authors mentioned mitofusins among dysregulated proteins but did not show any picture of the mitochondrial network in any models.

Minor points

line 41: genes are mutated rather than proteins

l 91-92: some Greek letters are missing in the pdf

Fig. 5: too many figures; panel E too small; what is AO in panel F?

Round 2

Reviewer 1 Report

The authors have addressed all of my concerns and I recommend the acceptance of the manuscript. There are still some grammar and typo errors that need to be fixed. For instance:

line 71: “this is the case” change to “as is the case”

line 99: “plays a critical role for the” change to “plays a critical role in the”

line 104: “proteases activity” change to “protease activity”

line 110: “which is part of’ change to “which is a part of”

line 483: “to evaluate strep one” change to “to evaluate step one”

line 487: “interacting with mitochondria, which we evaluated by analysis of colocalization ” change to “co-localizing with mitochondria”

Author Response

We would like to thank the reviewer for his/her useful suggestions and feedback. We have now corrected the typos and grammar as suggested.

Reviewer 2 Report

The authors replied properly to almost all my concerns.

While the absence of antibodies working on flies is a plausible explanation for some missing data, mtDNA assessment should be easily feasible.
